# Network Pharmacology Analysis of the Potential Pharmacological Mechanism of a Sleep Cocktail

**DOI:** 10.3390/biom14060630

**Published:** 2024-05-27

**Authors:** Yuyun Liang, Yanrong Lv, Jing Qin, Wenbin Deng

**Affiliations:** School of Pharmaceutical Sciences (Shenzhen), Shenzhen Campus of Sun Yat-sen University, Shenzhen 518107, China; liangyy56@mail3.sysu.edu.cn (Y.L.); lvyr8@mail.sysu.edu.cn (Y.L.)

**Keywords:** insomnia, sleep, network pharmacology, molecular docking, dietary supplement

## Abstract

Insomnia, also known as sleeplessness, is a sleep disorder due to which people have trouble sleeping, followed by daytime sleepiness, low energy, irritability, and a depressed mood. It may result in an increased risk of accidents of all kinds as well as problems focusing and learning. Dietary supplements have become popular products for alleviating insomnia, while the lenient requirements for pre-market research result in unintelligible mechanisms of different combinations of dietary supplements. In this study, we aim to systematically identify the molecular mechanisms of a sleep cocktail’s pharmacological effects based on findings from network pharmacology and molecular docking. A total of 249 targets of the sleep cocktail for the treatment of insomnia were identified and enrichment analysis revealed multiple pathways involved in the nervous system and inflammation. Protein–protein interaction (PPI) network analysis and molecular complex detection (MCODE) analysis yielded 10 hub genes, including AKT1, ADORA1, BCL2, CREB1, IL6, JUN, RELA, STAT3, TNF, and TP53. Results from weighted correlation network analysis (WGCNA) and Kyoto Encyclopedia of Genes and Genomes (KEGG) pathway enrichment analysis of insomnia-related transcriptome data from peripheral blood mononuclear cells (PBMCs) showed that a sleep cocktail may also ease insomnia via regulating the inflammatory response. Molecular docking results reveal good affinity of Sleep Cocktail to 9 selected key targets. It is noteworthy that the crucial target HSP90AA1 binds to melatonin most stably, which was further validated by MD simulation.

## 1. Introduction

Insomnia is a common subjective sleep disorder characterized by difficulty initiating and maintaining sleep as well as poor sleep quality, which results in weariness and discomfort upon waking [1]. As the pace of life in modern society accelerates and pressure increases, insomnia has become a very common and troublesome condition [2]. It has been estimated that insomnia affects up to 20% of the population, placing a significant burden on patients’ health and finances [3]. Sleep disturbances are associated with daytime functional deficits and increased fatigue levels, as well as an increased risk of numerous age-related diseases, including cardiovascular disease, stroke, diabetes, major depression, and dementia [4]. Benzodiazepines, alprazolam, and other sedative hypnotics are commonly prescribed to alleviate insomnia problems [5]. They can shorten the time it takes to fall asleep or wake up and lengthen the duration of sleep. Nevertheless, long-term use can lead to drug resistance and dependence as well as negative side effects such as fatigue, drowsiness, and cognitive decline [5]. Therefore, more and more people with insomnia are seeking alternative therapeutic options [6].

Many dietary supplements are used to improve sleep quality, and we observed several interesting ingredients: (1) 5-hydroxytryptophan (5-HTP, Figure 1A) is a precursor that is converted to serotonin (also known as 5-hydroxytryptamine or 5-HT), which plays an important role in regulating mood, anxiety, depression, and insomnia as well as in regulating regular body processes [7]. This conversion is facilitated by the enzyme aromatic-L-amino-acid decarboxylase and vitamin B6 [8]. (2) Melatonin (Figure 1B), secreted in the brain during the night, regulates the sleep–wake cycle in vertebrates and is used for managing sleep disorders [9]. (3) l-theanine (Figure 1C), structurally similar to the excitatory neurotransmitter glutamate, binds to glutamate receptors and is marketed for cognitive performance, stress reduction, improved sleep quality, and alleviating menstrual cramps [10]. (4) γ-Aminobutyric acid (GABA, Figure 1D) is the chief inhibitory neurotransmitter in mature mammalian central nervous systems [11]. GABA supplements can have the benefit of increasing GABA levels, thereby optimizing sleep quality and cognitive function and making people feel more secure and focused [12,13,14]. (5) Vitamin B6 (Figure 1E) serves as a cofactor in the biosynthesis of five important neurotransmitters: serotonin, dopamine, epinephrine, norepinephrine, and GABA. This ensures the physiological function of the nervous system [15,16]. (6) Magnesium is a vital component of nucleic acid chemistry in all known living organisms and there is growing evidence to suggest that magnesium supplementation can enhance sleep and relaxation [17,18]. The six supplements appear to function in a complementary manner.

A number of studies have demonstrated that the efficacy of these supplements is enhanced when they are used in conjunction with one another. Suhyeon Kim et al. found that GABA/l-theanine mixture has a positive synergistic effect on sleep quality and duration as compared to the GABA or l-theanine alone [19]. Studies in both invertebrates and vertebrates have found that GABA/5-HTP mixture regulated the sleep duration and increased the sleep quality more than single administration [20,21,22]. Gorica Djokic et al. found that magnesium–melatonin–vit B complex supplementation reduces insomnia symptoms [23]. Research from Carmela Bravaccio et al. demonstrated that the addition of tryptophan and vitamin B6 appears to have stronger influence on night awakenings reduction than melatonin only [24]. A review of existing research and the functions of these six supplements indicates that a combination of them has the potential to become a sleep cocktail for relieving insomnia. Furthermore, research indicates that some of these ingredients possess additional physiological mechanisms that may enhance sleep quality in addition to their effects on the nervous system. For instance, 5-HTP was able to improve the gut microbiota composition in poor sleepers [25]. GABA has been shown to regulate secretion of a greater number of cytokines and can suppress inflammatory immune responses [26,27,28]. This sleep cocktail involves multiple components and targets, and we employed network pharmacology methods [29] to uncover the network relationship between drug-gene–target–disease interactions, predict the mechanism of drug action through network relationships, and evaluate drug efficacy.

In this study, we aimed to systematically identify the molecular mechanisms of a sleep cocktail’s pharmacological effects based on findings from network pharmacology and molecular docking. The potential targets of the sleep cocktail’s active components were screened and anticipated, and the targets relevant to sleep were carefully analyzed. Then, utilizing network pharmacological approaches, the relationship between medications and targets was investigated. Weighted correlation network analysis (WGCNA) and Kyoto Encyclopedia of Genes and Genomes (KEGG) pathway enrichment analysis of insomnia-related transcriptome data from peripheral blood mononuclear cells (PBMCs) were also conducted. Molecular docking was employed to examine the interaction between the active compounds in the sleep cocktail and the essential therapeutic targets of sleep disorders. This research may prove to be a valuable resource for subsequent studies in this field.

## 2. Materials and Methods

### 2.1. Candidate Therapeutic Targets of Sleep Cocktail

The six bioactive constituents of the sleep cocktail are 5-hydroxytryptophan, magnesium, vitamin B6, l-theanine, GABA, and melatonin. The chemical ID, molecular weight, and canonical smiles of the aforementioned bioactive constituents were obtained from PubChem [30] (https://pubchem.ncbi.nlm.nih.gov/, accessed on 22 February 2024). The canonical SMILES format files were imported into the Swiss Target Prediction [31] platform (http://www.swisstargetprediction.ch/, accessed on 22 February 2024) with the attribute set to “homosapiens” and then screened to identify potential targets for the predicted chemical composition. Potential targets were identified from the STITCH database [32] (http://stitch.embl.de/, accessed on 22 February 2024) and CTD database [33] (http://ctdbase.org/, accessed on 22 February 2024). The obtained targets were pooled, and duplicates were deleted, yielding the final 1221 potential targets. Disease ontology (DO) [34] was performed using R package DOSE 3.19 [35].

### 2.2. Gene Collection of Insomnia

To find insomnia-related targets, a search of the OMIM [36] (http://www.omim.org, accessed on 22 February 2024), GeneCards [37] (http://www.genecards.org, accessed on 22 February 2024), and CTD databases [33] (http://ctdbase.org/, accessed on 22 February 2024) were conducted using the keyword “insomnia”. The UniProt database [38] (http://www.UniProt.org/, retrieved on accessed on 23 February 2024) was utilized to correct the obtained targets. Thereafter, the targets obtained in each database were summarized and duplicates deleted.

### 2.3. Common Targets and Compound–Target Network Construction

VENNY 2.1 [39] (https://bioinfogp.cnb.csic.es/tools/venny/, accessed on 23 February 2024) was utilized to intersect the obtained sleep cocktail targets and insomnia targets, thereby identifying the probable targets of the sleep cocktail that exert an influence on insomnia. A compound–target network of common targets and active ingredients was constructed using Cytoscape 3.10.1 [40].

### 2.4. Protein–Protein Interaction (PPI), Network Construction, and Network Analysis

The obtained potential targets were imported into the STRING database [41] (https://string-db.org, accessed on 24 February 2024), with the biological species set to “Homosapiens”, and the minimal interaction threshold set to “highest confidence” (>0.9). The remaining options were set to default parameters. The protein interaction network diagram was obtained, and PPI data were saved in TSV format. The PPI data file was imported into Cytoscape 3.10.1 for visualization, and any free nodes were removed. Finally, the “NetworkAnalysis” tool was used to examine the network topology characteristics in order to obtain the PPI network containing topological information.

The TRRUST database [42] (http://www.grnpedia.org/trrust, accessed on 5 March 2024) was employed to predict transcription factors (TFs) for regulatory hub genes. The adjusted *p* value was set at <0.05, which was considered significant. A target tissue/cell enrichment analysis of common targets was conducted utilizing the PaGenBase database [43] (http://bioinf.xmu.edu.cn/PaGenBase/, accessed on 5 March 2024).

To evaluate and select hub genes, Cytoscape’s cytoHubba [44] tool was employed to calculate the top 20 targets using the Degree, Maximum Neighborhood Component (MNC), Maximal Clique Centrality (MCC), and Closeness algorithms. The intersecting targets were defined as the central targets.

The Metascape platform [45] (http://metascape.org, accessed on 5 March 2024) was employed to analyze the biological process of interaction between the common targets and to identify the top functional modules using molecular complex detection (MCODE) [46], which uses a vertex-weighting scheme to discover local, high-density areas in the PPI graph.

### 2.5. GO and KEGG Enrichment Analysis and Compound–Target-Pathway Network Construction

The Database for Annotation, Visualization, and Integrated Discovery [47] (DAVID, https://david.ncifcrf.gov/, accessed on 6 March 2024) was employed to conduct gene ontology (GO) [48] and Kyoto Encyclopedia of Genes and Genomes (KEGG) [49] enrichment analyses. The top 10 GO enrichments and top 20 KEGG pathways were selected for further investigation, with a corrected *p*-value of ≤0.05. The results were subsequently visualized using the website https://www.bioinformatics.com.cn/ (accessed on 6 March 2024).

A compound–target pathway was constructed and analyzed in Cytoscape 3.10.1 to illustrate and elucidate the intricate relationships between active compounds, the top 20 enriched KEGG pathways, and targets in order to facilitate a more comprehensive understanding of the regulatory mechanisms of active compounds against insomnia.

### 2.6. Weighted Correlation Network Analysis

The GEOquery [50] R package was employed to download the GSE208668 [51] file from the GEO database (http://www.ncbi.nlm.nih.gov/geo/, accessed on 20 March 2024). The data set comprises whole transcriptome data from peripheral blood mononuclear cells (PBMCs) of 25 patients with insomnia and 17 control individuals. The present study utilized the data from 16 insomnia patients and 17 control individuals.

The WGCNA [52] R package was employed to identify the optimal soft threshold for the construction of a scale-free co-expression network. The construction of an adjacency matrix between gene expression profiles, the conversion of this matrix into a topological overlap matrix (TOM), the generation of a hierarchical gene clustering tree, and the identification of gene co-expression modules are the subsequent steps in this process. Finally, the relationship between each gene co-expression module and the groups was calculated. The genes within the module with the strongest correlation with the insomnia group were selected for subsequent analysis.

### 2.7. Molecular Docking

Molecular docking was performed on five compounds and nine target proteins selected based on PPI network analysis, MCODE analysis and KEGG results. The chemical structures of 5-hydroxytryptophan (CID_144), vitamin B6 (CID_1054), l-theanine (CID_439378), GABA (CID_119), and melatonin (CID_896) were downloaded from the PubChem database [30] (https://pubchem.ncbi.nlm.nih.gov/, accessed on 22 February 2024). The protein structures of target proteins were obtained from the PDB database [53] (https://www.rcsb.org/, accessed on 10 March 2024): ADORA1 (PDBID: 5uen), AKT1 (PDBID:7nh5), BCL2(PDBID:8hoi), FOS (PDBID:1s9k), HSP90AA1 (PDBID: 3o0i), JUN (PDBID:6emh), MAPK3 (PDBID:4qtb), TNF (PDBID: 2az5), TP53 (PDBID: 6mxy).

PyMOL [54] 2.6.0 was employed to extract water molecules, original ligands, and peptides from the target protein file. AutoDock Tools 1.5.6 [55] was then used to import and process the target protein for hydrogenation, charge calculation, and nonpolar hydrogen combination. Subsequently, the processed protein structure is then exported using the PDBQT format. The active pocket location is constructed based on the position of the original ligand or original peptide in the three-dimensional structure. The size of the grid box and the X, Y, and Z centers were then adjusted accordingly [56]. AutoDock Vina 1.1.2 [57] was employed to conduct semi-flexible docking and to identify optimal conformations. The results subjected to a comprehensive investigation and visualization process, utilizing PyMOL 2.6.0, LigPlus v.2.2.8 [58], and PLIP tools [59] (https://plip-tool.biotec.tu-dresden.de/plip-web/plip/, accessed on 15 March 2024).

### 2.8. Molecular Dynamics Simulation

Molecular dynamics (MD) simulation was performed to simulate the binding stability of Melatonin and HSP90AA1 using Gromacs [60] 2019.6 at constant temperature and pressure as well as periodic boundary conditions. The amber99sb-ildn protein force field, GAFF force field, and TIP3P water model were applied. Small fractions were constrained during MD simulations using the LINCS algorithm for all involved hydrogen bonds with an integration step of 2 fs. Electrostatic interactions were calculated using the particle mesh Ewald (PME) method. The non-bonding interaction cutoff was set to 12 Å and updated every 20 steps. The V-rescale method was used to control the simulation temperature to 300 K, and the Parrinello–Rahman method was used to control the pressure to 1 bar.

First, the steepest descent method was used to minimize the energy of the system in order to eliminate the too-close contact between the atoms; then, all the atoms of the system were heated up to 300 K within 100 ps; finally, the system was subjected to a 100 ns MD. Finally, the system was subjected to 100 ns MD simulations, and the conformations were saved every 20 ps. In order to further investigate the properties of the system, the MM-PBSA method was used for free energy calculation and decomposition to gain insight into the mechanical basis of protein–small molecule interactions.

## 3. Results

### 3.1. Common Targets between Sleep Cocktail and Insomnia and PPI Network Construction

The sleep cocktail contains the following ingredients: 5-hydroxytryptophan, magnesium, vitamin B6, l-theanine, GABA, and melatonin. A total of 1219 potential targets were identified using the Swiss Target Prediction platform [31], STITCH database [32], and CTD database [33] (Appendix A). To gain further insight into the disease connections of the ingredients, a disease ontology (DO) enrichment analysis was conducted. The targets of these ingredients are enriched in diseases such as ischemia, brain ischemia, and kidney failure, and many are related to the nervous system (Appendix A).

A total of 876 insomnia-related disease targets were obtained after collecting, screening and removing duplicate targets from CTD [33], Gene Cards [37] and OMIM database [36]. A total of 249 common targets were identified by intersecting the 1219 potential sleep cocktail targets with the 876 insomnia disease gene sets (Figure 2A). The sleep cocktail component–target interaction network was constructed using Cytoscape 3.10.1 [40], as shown in Figure 2B.

The PaGenBase database [43] was utilized to conduct target tissue/target cell enrichment analyses on common targets. Some of them were selectively expressed in the caudate nucleus, lung, and dorsal root ganglion (DRG) (Figure 2C). A transcription factor (TF)–target analysis of the TRRUST database [42] reveals that numerous targets are regulated by TFs such as NFKB1, RELA, SP1, JUN, and others, suggesting that these TFs may play an important role in insomnia (Figure 2D).

In order to investigate the mechanism of the sleep cocktail on insomnia, 249 targets were imported into the STRING database [41] to form a protein–protein interaction (PPI) network (Figure 3A). After the removal of nodes that are not connected to others, the PPI network consists of 244 nodes and 6005 edges, with an average node degree of 49.2 and a local clustering coefficient of 0.599. The 10 most highly connected nodes in the network are AKT1 (152), ACTB (150), INS (150), ALB (148), TNF (142), IL1B (139), IL6 (137), BDNF (130), CREB1 (127), CASP3 (126), and TP53 (126). The different colored lines between edges indicate the type of interaction evidence. The network was imported into Cytoscape 3.10.1 and the NetworkAnalyzer tool 4.5.0 was utilized to examine the topological network (Figure 3B). The color of the node becomes darker as the degree of the node increases, and the size of the node becomes larger as the betweenness of the node increases. The larger nodes and darker colors indicate a greater biological significance associated with insomnia.

### 3.2. GO, KEGG Pathway Enrichment Analysis, and Compound–Target Pathway Network

We then explored the core functions and pathways of 249 potential targets through gene ontology [48] (GO) and Kyoto Encyclopedia of Genes and Genomes [49] (KEGG) pathway enrichment analysis. Based on the GO enrichment analysis of biological processes (BP), cellular composition (CC), and molecular function (MF) (Appendix A), the top 10 with the lowest *p*-values in each category were selected and visualized (Figure 4A). The biological processes were primarily associated with responses to xenobiotic stimulus, chemical synaptic transmission, circadian rhythm, and negative regulation of neuron apoptotic process. The cellular components were mainly distributed in nerve-related cell composition, synapse, neuronal cell body, dendrite, postsynaptic membrane, and so forth. The molecular function mainly involved identical protein binding, enzyme binding, protein binding, neurotransmitter receptor activity. Many genes were enriched in terms related to the nervous system, as expected, and others suggested further biological functions of these targets.

Furthermore, 184 KEGG pathways (Appendix A) were identified through enrichment analysis and the top 20 highly enriched pathways are shown in Figure 4B,C. These targets are enriched in Lipid and atherosclerosis, Neuroactive ligand-receptor interaction, Pathways of neurodegeneration—multiple diseases and other pathways. Using Cytoscape 3.10.1, we constructed a compound–target pathway network containing the top 20 KEGG pathways, their linked targets and sleep cocktail components (Figure 4D), which consists of 277 nodes and 1798 edges. The top 5 targets are involved in 17 (MAPK8, MAPK9, NFKB1, RELA) and 16 (AKT1) pathways. The substantial enrichment of the TNF signaling pathway and the IL-17 signaling pathway suggests that sleep cocktail may play a role in the treatment of insomnia through these two pathways, and both pathways are related to the inflammatory response (Appendix A).

### 3.3. PPI Network Hub Genes and Functional Cluster Analysis

The hub genes of the PPI network were identified using the CytoHubba [44] tool in Cytoscape. The top 20 hub genes from each of the five commonly used algorithms—Degree, Maximum Neighborhood Component (MNC), Maximal Clique Centrality (MCC), and Closeness—were calculated (Figure 5A–E). Nine common hub genes were identified (Table 1) by intersection (Figure 5F). GO enrichment analysis showed that these genes mainly constitute the transcription factor complex and are involved in the regulation of transcription (Appendix A). KEGG pathway analysis showed that the hub genes were mainly involved in various virus infections, apoptosis, etc. (Appendix A).

In addition, the molecular complex detection (MCODE) [46] method was applied to this network using Metascape [45] to discover significant modules of densely protein-linked areas. In total, 11 modules consisting of 122 targets were obtained (Figure 6A) and visualized using Cytoscape (Figure 6B–L). MCODE 1 and MCODE 3 were regarded as important modules as their MCODE scores were higher than 4. With seed node IL6, MCODE 1 comprises 38 nodes and 303 edges (Figure 6B, score = 7.97). The proteins in this cluster are associated with immune function-related processes such as interleukin-4 and interleukin-13 signaling, the AGE-RAGE signaling pathway in diabetic complications, and signaling by interleukins according to GO enrichment analysis. MCODE 3 has 20 nodes and 153 edges (Figure 6D, score = 7.65), with seed node ADORA1. GO analysis reveals that the proteins in the cluster bind to GPCR ligand, G alpha (i) signaling events, and Class A/1 (Rhodopsin-like receptors) (Appendix A), which are involved in the conversion of extracellular signals into key physiological effects.

### 3.4. Weighted Correlation Network Analysis and Functional Analysis of PBMC Transcriptome

Studies have shown reciprocal connections between the central nervous system, sleep, and the immune system. To investigate whether the sleep cocktail is involved in inflammation-related pathways, we downloaded GSE208668 [51] from the Gene Expression Omnibus datasets (GEO: www.ncbi.nlm.nih.gov/geo/, (accessed on 20 March 2024)) and performed weighted correlation network analysis (WGCNA) on the peripheral blood mononuclear cell (PBMC) transcriptome data of 16 insomnia and 17 normal groups to find co-expressed gene modules and explore the association between the gene network and the phenotype of interest, as well as the core genes in the network. Through WGCNA analysis, 5000 genes were divided into 9 gene modules (Figure 7A), and the Pearson correlation coefficients between the modules and groups were calculated (Figure 7B). The turquoise module showed a correlation of cor = 0.98 with the normal group, while the blue module had a correlation of cor = 0.9 with the insomnia group. We then measured gene significance (GS) for each gene’s traits and the module membership (MM) in the modules, showing that genes in these two modules have substantial relationships with traits compared to other modules (Figure 7C,D and Appendix A).

According to the KEGG enrichment results of these two modules (Figure 8), they were co-enriched on the pathways of Huntington disease, oxidative phosphorylation, prion disease, non-alcoholic fatty liver disease, amyotrophic lateral sclerosis, Parkinson disease, and RNA polymerase. In addition to the common enriched pathways, the turquoise module is also enriched in Epstein–Barr virus infection, nucleotide excision repair, Kaposi-sarcoma-associated herpesvirus infection, and others. We compared these findings with those of our previously identified targets and found that they were co-enriched in the pathways of Huntington disease, oxidative phosphorylation, prion disease, amyotrophic lateral sclerosis, Parkinson disease, pathways of neurodegeneration—multiple diseases and multiple neurological-related diseases (Appendix A). This dataset was derived from PBMCs, but the enriched results reveal certain neurological-disease-related pathways, demonstrating the complicated relationship between inflammatory response and the nervous system. The genes in the turquoise and blue modules do not overlap, implying that genes from two modules with opposite correlations may play opposing roles in the same gene pathway. Common target genes have only minor overlap with module genes (more from the turquoise module, Appendix A), suggesting that sleep cocktail may influence insomnia-related inflammatory responses through these pathways.

### 3.5. Molecular Docking Verification

Assessment of the binding mechanisms between compounds and disease-associated targets is enabled by the application of molecular docking technologies. Based on the PPI network, MCODE and KEGG results, nine insomnia-related targets were selected: ADORA1, AKT1, BCL2, FOS, HSP90AA1, JUN, MAPK3, TNF, TP53, and molecular docking was performed with five compounds: 5-HTP, vitamin B6, l-theanine, GABA, and melatonin. The binding energy (kcal/mol) was obtained via Autodock Vina 1.1.2 [57] docking analysis, and the results are presented in Figure 9 and Table 2. Lower values indicate a stronger binding ability, and binding scores less than −5.0 kcal/mol are generally considered to indicate medium affinity, while binding fractions less than −7.0 kcal/mol are considered to indicate high affinity. The binding affinities of nine complexes are lower than −7.0 kcal/mol (HSP90AA1-melatonin, AKT1-5-HTP, HSP90AA1-5-HTP, TP53-melatonin, MAPK3-5-HTP, MAPK3-melatonin, AKT1-melatonin, TNF-5-HTP, TP53-5-HTP), indicating that the compounds are stable in binding to the protein targets.

The interactions of the docking complexes were obtained using PLIP [59] (https://plip-tool.biotec.tu-dresden.de/plip-web/plip/, accessed on 15 March 2024) and are presented in Table 2. The three-dimensional graph of the compound–target interaction complex with the highest free binding energy for each target or compound was visualized using PyMOL [54] 2.6.0, as shown in Figure 10. LigPlus 2.2.8 [58] was used to generate 2D plots for molecular docking visualization (Appendix A). The active site residues of the compounds and the compounds themselves were found to be involved in hydrophobic interaction, hydrogen bonding, pi stacking (parallel), pi cation interaction, and salt bridge.

Molecular dynamics (MD) simulation has found extensive application in assessing the structural attributes of protein–ligand systems and investigating the enduring bond between proteins and molecules [61]. In this study, the crucial target HSP90AA1 was chosen to further examine the stability of binding to melatonin. The RMSD values of the protein backbone exhibited fluctuations between 0.09 nm and 0.15 nm throughout MD simulations (Figure 11A). The stabilized RMSD values of backbone atoms confirm that the system is well balanced. During the simulation, 0–1 hydrogen bonding interactions were formed between proteins and small molecules (Figure 11B). Binding free energy was calculated for the last 10 ns using the MM-PBSA method (Figure 11C). The results indicate that van der Waals forces are the primary energy contributor during small molecule–protein interactions with a binding energy of −127.130 +/− 1.021 kJ/mol. Additionally, electrostatic interactions (−17.796 +/− 0.804 kJ/mol) and nonpolar solvation energies (−14.964 +/− 0.111 kJ/mol) are beneficial for the binding of small molecules and proteins. Although the solvation energy (79.428 +/− 1.192 kJ/mol) acts as a barrier to protein–small molecule binding, the high van der Waals forces counteract, resulting in a protein–ligand interaction energy of −80.306 +/− 1.504 kJ/mol. The free energy was decomposed in order to explore the contribution of amino acids to substrate binding energy. The top 10 amino acid sites contributing to protein-molecule binding were identified as residue 107, 138, 54, 47, 98, 162, 111, 150, 93, and 108 (Figure 11D, Appendix A).

## 4. Discussion

Although these ingredients have been studied for their sleep benefits in recent years, most research has focused on single ingredients, while most dietary supplements on the market are combinations. However, there has been limited research exploring the mechanisms of dietary supplement combinations. This study is the first to use network pharmacology and molecular docking methods to systematically analyze the molecular mechanisms of a sleep cocktail (5-hydroxytryptophan, magnesium, vitamin B6, l-theanine, GABA, and melatonin) in the treatment of insomnia.

The findings from enrichment analysis and network analysis suggest that the sleep cocktail may regulate insomnia symptoms by participating in signal transduction-related pathways of the nervous and immune systems. Studies have shown reciprocal links between sleep and the immune system [62]. Increased systemic inflammation, which has been found to mediate mortality risk, is prospectively predicted by both self-reported sleep disturbance and objective short sleep duration [63]. In healthy adults, partial sleep deprivation for one night has been shown to increase the production of pro-inflammatory cytokines, including IL-6 and tumor necrosis factor (TNF)-α, by monocytes [64,65,66]. This can activate transcriptional control pathways of inflammation, such as nuclear factor (NF)-κB/Rel family molecules [66,67]. Notably, the identified core targets (e.g., TP53, TNF, STAT3, RELA, JUN, IL6, CREB1, BCL2, and AKT1) and enriched pathways (e.g., TNF signaling pathway, IL-17 signaling pathway) align with previous studies linking sleep disturbances to increased systemic inflammation and the production of pro-inflammatory cytokines WGCNA analysis further supports the potential role of the sleep cocktail in modulating the immune system, as key modules associated with insomnia were enriched for similar pathways identified in this study.

Molecular docking simulations revealed stable binding interactions between the sleep cocktail compounds and key targets like HSP90AA1, AKT1, and TP53. These proteins are implicated in various cellular processes, including stress adaptation, cell signaling, and apoptosis regulation, which could contribute to the sleep-promoting effects of the cocktail. The AKT1 protein, also known as protein kinase B (PKB), is a key molecule in cell signaling pathways and is involved in regulating various processes such as cell growth, survival, metabolism, and differentiation [68]. Current research on HSP90AA1 focuses on its role as a drug target due to its interaction with a variety of tumor-promoting proteins and its role in cellular stress adaptation, including AKT1 [69]. In addition, extracellular Hsp90A induces inflammation through activation of the NF-κB (RELA) and STAT3 transcriptional programs, including the proinflammatory cytokines IL-6 and IL-8 [70,71]. The five compounds all have good affinity to key targets, such as TP53 (melatonin: −7.7; 5-HTP: −7.1; l-theanine: −6.3; vitaminB6: −5.7; GABA: −4.8). The p53 protein expressed by this gene coordinates multiple responses, including cell cycle arrest, DNA repair, metabolic changes, antioxidant effects, anti-angiogenic effects, autophagy, senescence, and apoptosis [72]. The strong binding affinities suggest that the cocktail components may exert their effects by modulating these targets and associated pathways.

This study provides a comprehensive analysis of potential targets and mechanisms underlying the sleep cocktail, which could guide the development of novel insomnia treatments. However, it should be noted that the target data were derived from databases, and the reliability and accuracy of the analysis and predictions depend on the quality of the data. Additionally, further in vivo and in vitro studies are needed to validate the results and elucidate the specific roles of each component in the sleep cocktail.

## 5. Conclusions

This study employed a network pharmacology approach to investigate the potential mechanisms underlying the sleep cocktail—a combination of dietary supplement ingredients (5-hydroxytryptophan, magnesium, vitamin B6, l-theanine, GABA, and melatonin)—in the treatment of insomnia. The analysis identified key targets, pathways, and processes that the sleep cocktail may modulate, including inflammatory response pathways (e.g., TNF signaling, IL-17 signaling), signal transduction pathways in the nervous and immune systems, and processes related to cell growth, survival, and apoptosis. Molecular docking simulations further revealed the stable binding affinities of the compounds with crucial targets, including HSP90AA1, AKT1, and TP53, suggesting their potential roles in mediating the sleep-promoting effects. The crucial target HSP90AA1 binds to melatonin most stably, which was further validated by MD simulation.

## Figures and Tables

**Figure 1 biomolecules-14-00630-f001:**
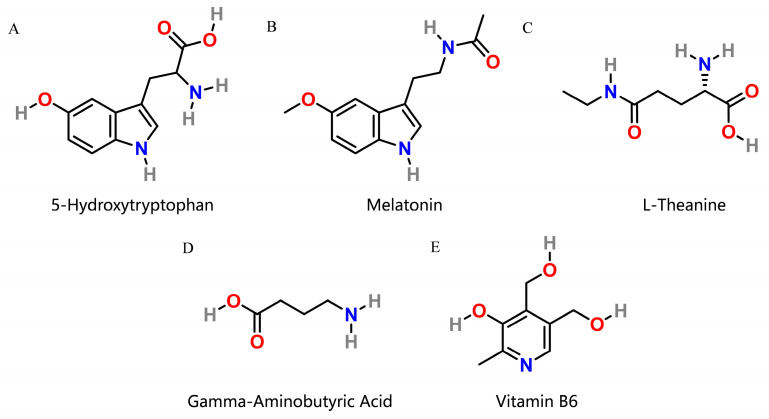
Two-dimensional structures of 5-hydroxytryptophan (5-HTP), melatonin, l-theanine, γ-aminobutyric acid (GABA) and vitamin B6. (**A**) 2D structure of 5-HTP. (**B**) 2D structure of melatonin. (**C**) 2D structure of l-theanine. (**D**) 2D structure of GABA. (**E**) 2D structure of vitamin B6.

**Figure 2 biomolecules-14-00630-f002:**
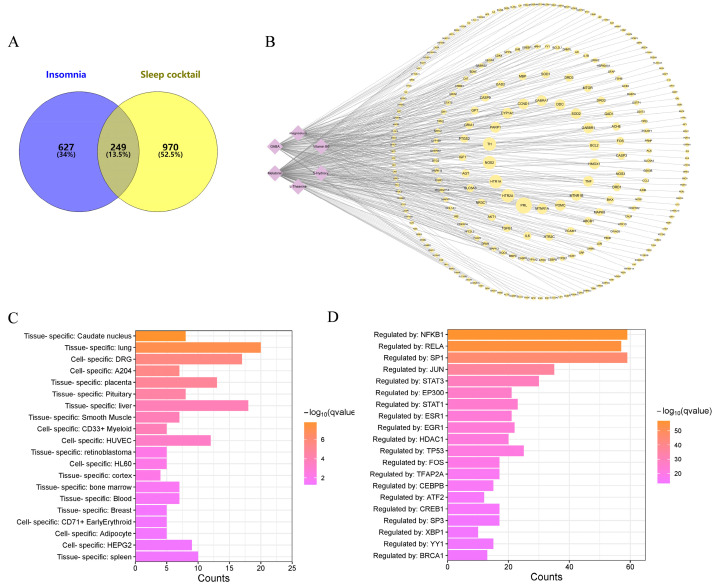
Common target genes of sleep cocktail treating insomnia. (**A**) Venn diagram of target genes in both sleep cocktail targets and insomnia genes collection. (**B**) Compound–target network. (**C**) Histogram of enrichment analysis of common target genes in PaGenBase. (**D**) Transcription factor (TF) enrichment analysis of common target genes in TRRUST.

**Figure 3 biomolecules-14-00630-f003:**
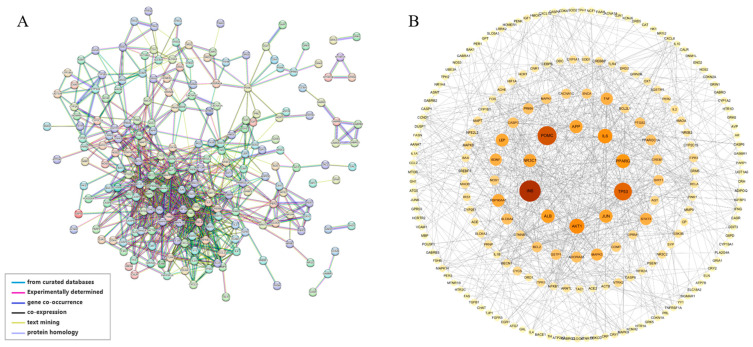
PPI network from (**A**) STRING database and (**B**) topological analysis.

**Figure 4 biomolecules-14-00630-f004:**
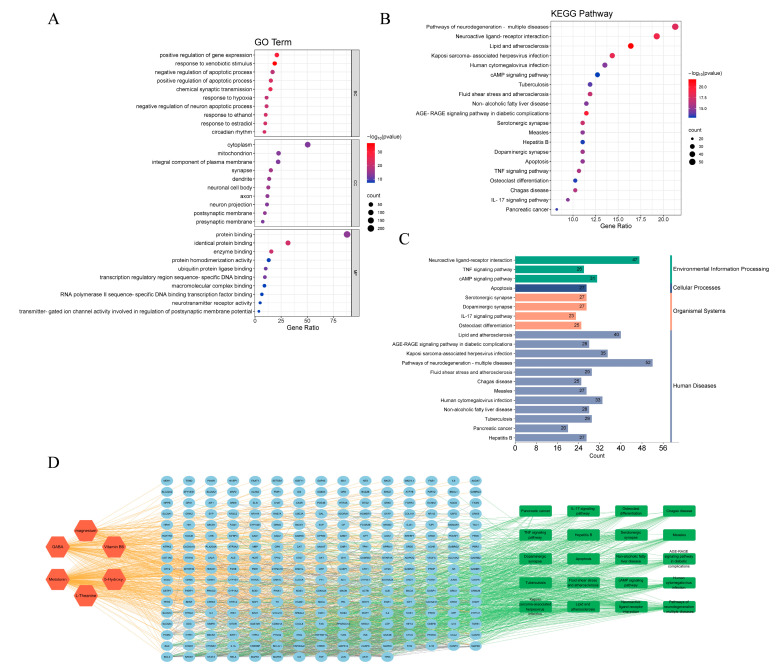
Gene ontology (GO) and Kyoto Encyclopedia of Genes and Genomes (KEGG) pathway enrichment analysis. (**A**) GO function analysis. (**B**) KEGG pathways enrichment analysis. (**C**) KEGG pathways classification. (**D**) Compound–target pathway network.

**Figure 5 biomolecules-14-00630-f005:**
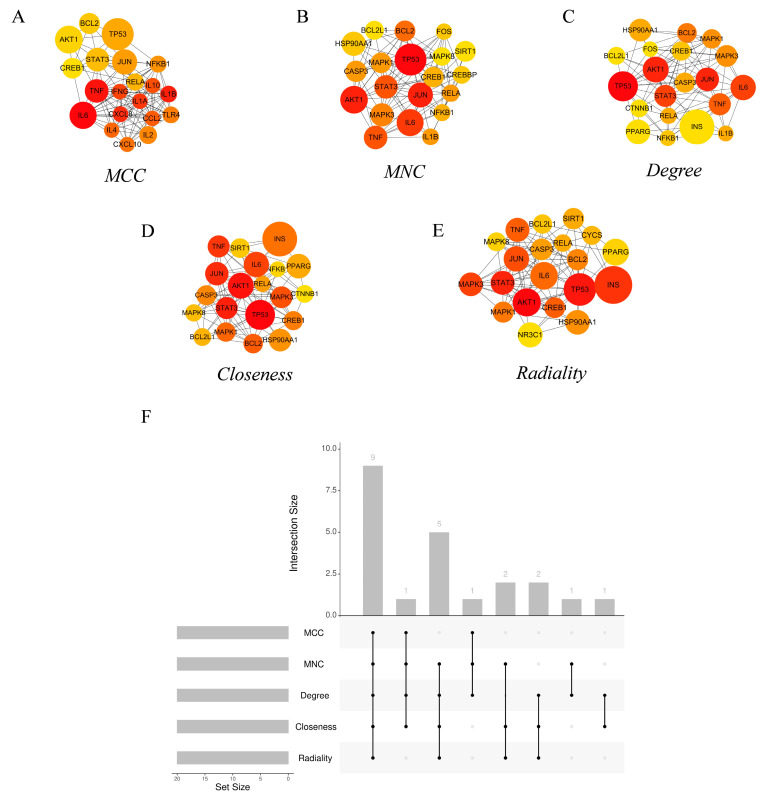
Core target networks identified by (**A**) MCC, (**B**) MNC, (**C**) Degree, (**D**) Closeness, and (**E**) Radiality. (**F**) Upset diagram of the common core targets of MCC, MNC, Degree, Closeness, and Radiality.

**Figure 6 biomolecules-14-00630-f006:**
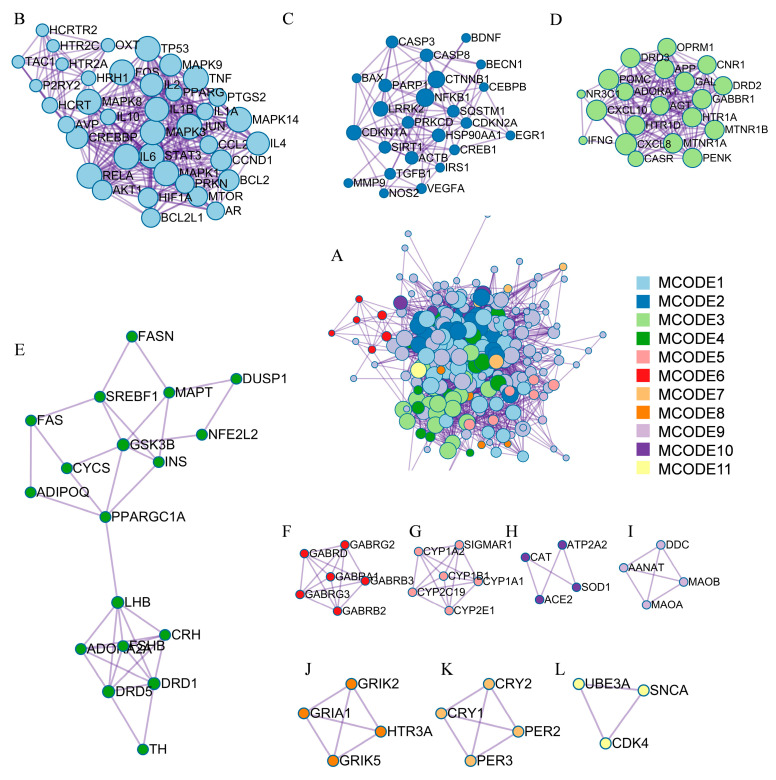
MCODE cluster analysis. (**A**) All MCODE clusters. (**B**) Targets cluster for MCODE1. (**C**) Targets cluster for MCODE2. (**D**) Targets cluster for MCODE3. (**E**) Targets cluster for MCODE4. (**F**) Targets cluster for MCODE5. (**G**) Targets cluster for MCODE6. (**H**) Targets cluster for MCODE7. (**I**) Targets cluster for MCODE8. (**J**) Targets cluster for MCODE9. (**K**) Targets cluster for MCODE10. (**L**) Targets cluster for MCODE11.

**Figure 7 biomolecules-14-00630-f007:**
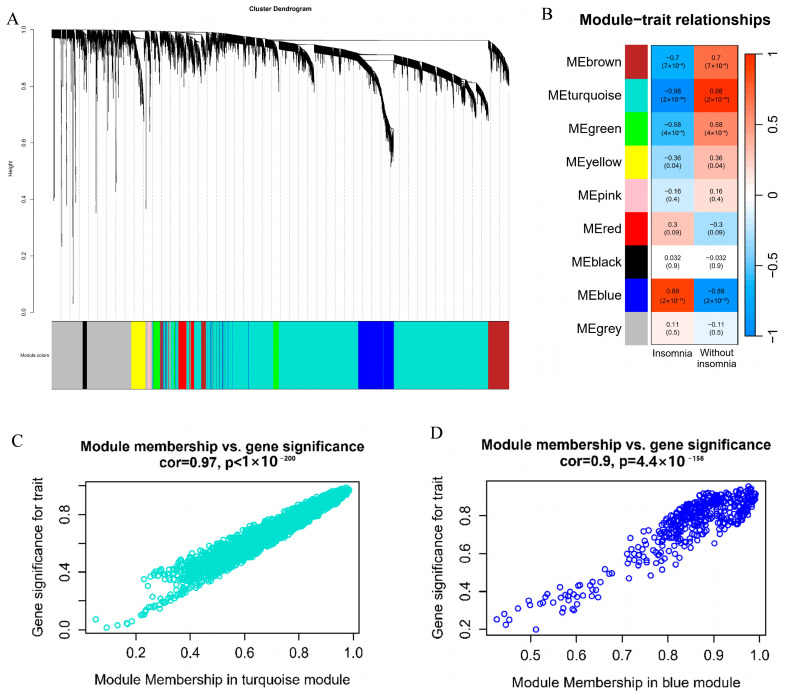
Weighted correlation network analysis (WGCNA) of peripheral blood mononuclear cell (PBMC) transcriptome data. (**A**) Gene dendrogram and modules; (**B**) Pearson correlation analysis of modules and groups; (**C**) scatterplot of module membership (MM) and gene significance (GS) from the turquoise module; (**D**) scatterplot of MM and GS from the blue module.

**Figure 8 biomolecules-14-00630-f008:**
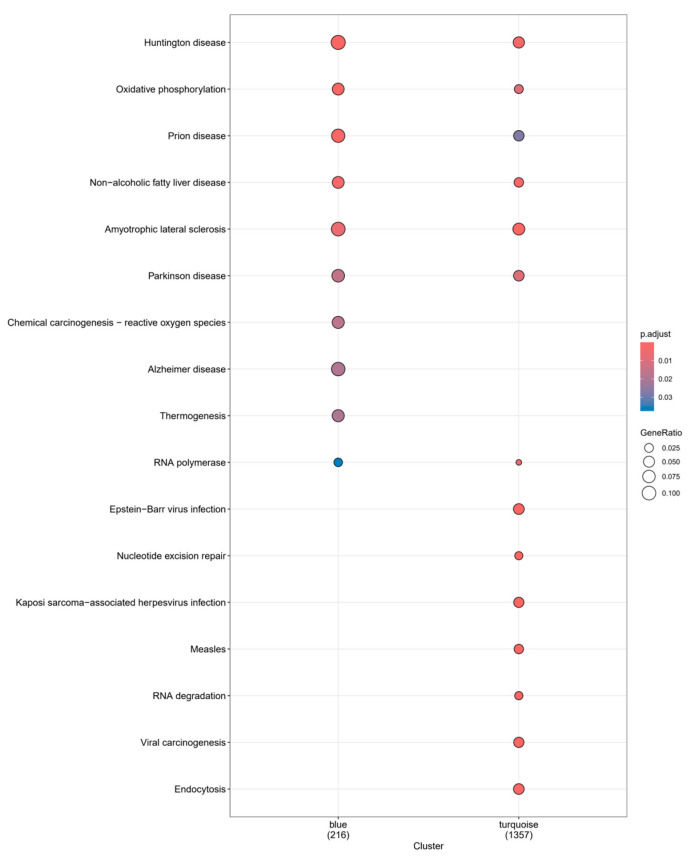
GO function analysis of turquoise and blue modules.

**Figure 9 biomolecules-14-00630-f009:**
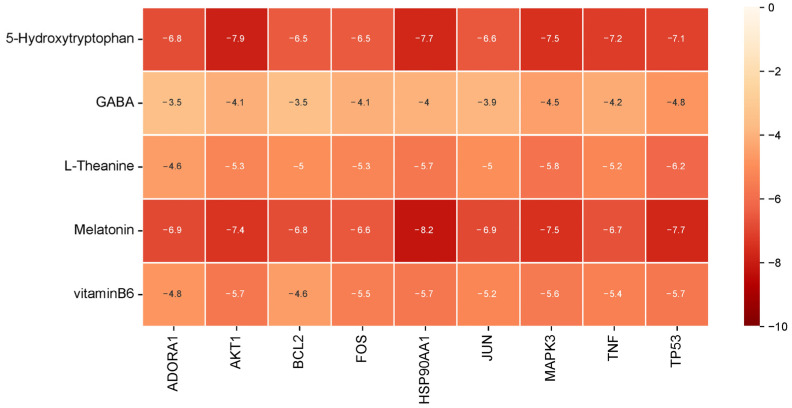
Heatmap of the binding energy (kcal/mol) of key targets and active compounds.

**Figure 10 biomolecules-14-00630-f010:**
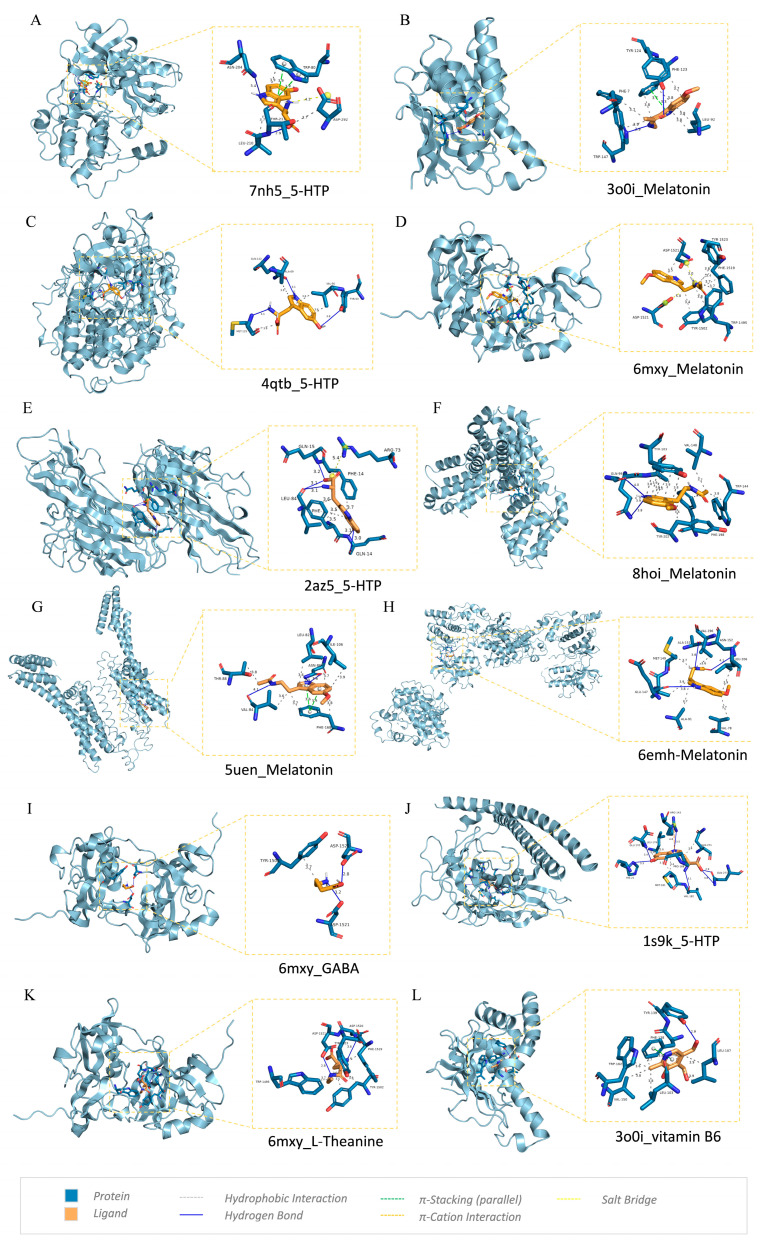
Molecular docking 3D diagram of key targets and active compounds: (**A**) 7nh5(AKT1)_5-HTP; (**B**) 3o0i(HSP90AA1)_Melatonin; (**C**) 4qtb(MAPK3)_5-HTP; (**D**) 6mxy(TP53)_Melatonin; (**E**) 2az5(TNF)_5-HTP; (**F**) 8hoi(BCL2)_Melatonin; (**G**) 5uen(ADORA1)_Melatonin; (**H**) 6emh_Melatonin; (**I**) 6mxy(TP53)_GABA; (**J**) 1s9k_5-HTP; (**K**) 6mxy(TP53)_L-Theanine; (**L**) 3o0i(HSP90AA1)_vitamin B6.

**Figure 11 biomolecules-14-00630-f011:**
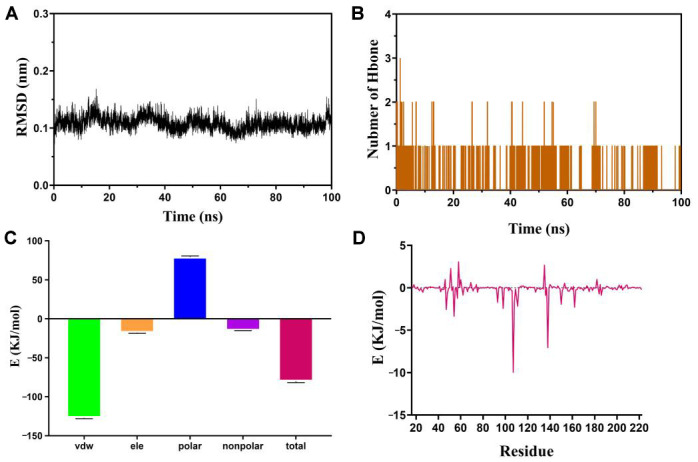
The MD simulation results of HSP90AA1 and melatonin. (**A**) RMSD plot during molecular dynamics simulations. (**B**) Number of hydrogen bones. (**C**) Binding free energy. (**D**) Contribution of the amino acids to the binding free energy.

**Table 1 biomolecules-14-00630-t001:** Hub genes identified by MCC, MNC, Degree, Closeness, and Radiality.

NO.	UniProt ID	Gene Symbol	Protein Name
1	P31749	AKT1	RAC-alpha serine/threonine-protein kinase, 2.7.11.1, protein kinase B, PKB, protein kinase B alpha, PKB alpha, proto-oncogene c-Akt, RAC-PK-alpha
2	P10415	BCL2	apoptosis regulator Bcl-2
3	P16220	CREB1	Cyclic AMP-responsive element-binding protein 1, CREB-1, cAMP-responsive element-binding protein 1
4	P05231	IL6	Interleukin-6, IL-6, B-cell stimulatory factor 2, BSF-2, CTL differentiation factor, CDF, hybridoma growth factor, Interferon beta-2, IFN-beta-2
5	P05412	JUN	Transcription factor Jun, activator protein 1, AP1, proto-oncogene c-Jun, transcription factor AP-1 subunit Jun, V-jun avian sarcoma virus 17 oncogene homolog, p39
6	Q04206	RELA	Transcription factor p65, Nuclear factor NF-kappa-B p65 subunit, nuclear factor of kappa light polypeptide gene enhancer in B-cells 3
7	P40763	STAT3	Signal transducer and activator of transcription 3, acute-phase response factor
8	P01375	TNF	Tumor necrosis factor, cachectin, TNF-alpha, tumor necrosis factor ligand superfamily member 2, TNF-a
9	P04637	TP53	Cellular tumor antigen p53, antigen NY-CO-13, phosphoprotein p53, tumor suppressor p53

**Table 2 biomolecules-14-00630-t002:** Table of the binding energy (kcal/mol) and interacting residues of key targets and active compounds.

Protein	Ligand	Binding Affinity	Interacting Residues
HSP90AA1	Melatonin	−8.2	TYR-124, PHE-123, PHE-7, TRP-147, LEU-92
AKT1	5-HTP	−7.9	ASN-204, THR-211, TRP-80, ASP-292, LEU-210
TP53	Melatonin	−7.7	ASP-1521, TYR-1502, TYR-1523, PHE-1519, TRP-1495
MAPK3	5-HTP	−7.5	MET-125, GLN-122, ALA-69, VAL-56, TYR-53
TNF	5-HTP	−7.2	GLN-15, ARG-73, LEU-84, PHE-14, PHE-14, GLN-14
ADORA1	Melatonin	−6.9	THR-88, LEU-82, ILE-106, ASN-86, PHE-168
JUN	Melatonin	−6.9	VAL-196, ASN-152, GLU-147, MET-149, LEU-206, ALA-91, VAL-78
BCL2	Melatonin	−6.8	GLN-99, TYR-103, VAL-148, TYR-202, PHE-198, TRP-144
FOS	Melatonin	−6.6	ARG-143, GLU-170, LEU-179, GLN-271, HIS-21, MET-181, PRO-168, VAL-182, GLN-273
TP53	L-Theanine	−6.2	ASP-1520, TRP-1495, ASP-1521, PHE-1519, TYR-1502
HSP90AA1	vitaminB6	−5.7	TYR-139, PHE-138, TRP-162, LEU-107, VAL-150, LEU-103
TP53	GABA	−4.8	ASP-1521, TYR-1502, ASP-1521

## Data Availability

The authors confirm that the data supporting the findings of this study are available within the article and its Appendix A.

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
