# Peer review of "Network Pharmacology Analysis of the Potential Pharmacological Mechanism of a Sleep Cocktail"

_biomolecules, 2024, doi:10.3390/biom14060630_

Round 1

Reviewer 1 Report

Comments and Suggestions for Authors

The authors of this manuscript applied an interesting computational workflow, including network analyses and molecular docking simulations, to identify possible molecular targets and pathways involved in the activity of a "sleep cocktail", composed of 5-Hydroxytryptophan, Magnesium, Vitamin B6, L-Theanine, GABA and Melatonin, which showed efficacy in counteracting insomnia.

The manuscript is well written and clear, and I think the workflow leading to the selection of potential targets is remarkable and well performed. My main concern involves the molecular docking simulations, since most of the obtained interaction energies are quite low. I assume that the authors used the rigid molecular docking procedure in AutoDock Vina, since it is not specified otherwise in the methods, but I would suggest running all simulations again using the flexible molecular docking procedure, which allows to approximate a dynamic adaptation of the residues surrounding the interaction site during ligand binding. I believe this would make the manuscript more suitable for publication, improving the computational results and providing a clearer understanding of the interaction of the compounds with the predicted targets.

Additionally, I have a few minor observations:

- page 4, line 171 - the common targets indicated are 249, while in Fig. 1A the common targets are 221

- I think fig. 4 would be better placed in the supplementary material

- I would also add a figure with the 2D structures of the compounds

- In the text and figures legends, it would be more clear to indicate the proteins by their name and not their PDB ID

Author Response

The authors of this manuscript applied an interesting computational workflow, including network analyses and molecular docking simulations, to identify possible molecular targets and pathways involved in the activity of a "sleep cocktail", composed of 5-Hydroxytryptophan, Magnesium, Vitamin B6, L-Theanine, GABA and Melatonin, which showed efficacy in counteracting insomnia.

The manuscript is well written and clear, and I think the workflow leading to the selection of potential targets is remarkable and well performed. My main concern involves the molecular docking simulations, since most of the obtained interaction energies are quite low. I assume that the authors used the rigid molecular docking procedure in AutoDock Vina, since it is not specified otherwise in the methods, but I would suggest running all simulations again using the flexible molecular docking procedure, which allows to approximate a dynamic adaptation of the residues surrounding the interaction site during ligand binding. I believe this would make the manuscript more suitable for publication, improving the computational results and providing a clearer understanding of the interaction of the compounds with the predicted targets.

-Thank you for your insightful comments and suggestions regarding our manuscript.

-Regarding your concern about the molecular docking simulations and the obtained interaction energies, we acknowledge your valid point. We used the semi-flexible molecular docking procedure in AutoDock Vina, and we revised in the method section (Revived main text, line 188). “AutoDock Vina [57] was employed to conduct semi-flexible docking and to identify optimal conformations.”

-Thank you again for your valuable suggestion regarding flexible molecular docking simulations. While we recognize the potential benefits of this approach and made sincere efforts to implement the flexible docking procedure as recommended, we are currently limited by the lack of sufficient computing resources required for these computationally intensive simulations. However, we acknowledge the importance of flexible docking simulations and will explore opportunities to access the necessary computing resources in the future.

Additionally, I have a few minor observations:

- page 4, line 171 - the common targets indicated are 249, while in Fig. 1A the common targets are 221"

Thank you for pointing out the discrepancy between the number of common targets mentioned in the text (249) and the number shown in Fig. 1A (221). The diagram in question was erroneous and has been replaced in the revised version. We also double-checked our data and updated the supplementary data.

"- I think fig. 4 would be better placed in the supplementary material"

Thank you for your comments. We have moved it to the appendix.

"- I would also add a figure with the 2D structures of the compounds"

We appreciate your suggestion and add a figure with the 2D structures of the compounds.

- In the text and figures legends, it would be more clear to indicate the proteins by their name and not their PDB ID

Thank you for the recommendation regarding the use of protein names instead of PDB IDs in the text and figure legends. We agree that this change and implemented this suggestion.

Reviewer 2 Report

Comments and Suggestions for Authors

Thank you for inviting me to review this manuscript. I read this manuscript with interest and care. In my opinion, the manuscript is suitable for Biomolecules publication after clarification of the following points:

1. The abstract need to contain more significant results and values to be more attractive.

2. The introduction need to be improved and more recent articles related to the subject should be added, 

3. The quality of the figures should be improved.

4. The English language should be checked carefully.

5. It will be goo if the authors can add molecular dynamics simulation.

6. The conclusion should be added.

Comments on the Quality of English Language

The English should be improved.

Author Response

Thank you for inviting me to review this manuscript. I read this manuscript with interest and care. In my opinion, the manuscript is suitable for Biomolecules publication after clarification of the following points:

  1. The abstract need to contain more significant results and values to be more attractive.

Thank you for your comments. We have revised the abstract as suggested.

  1. The introduction need to be improved and more recent articles related to the subject should be added,

Thank you for your comments. We agree that the introduction section should provide a comprehensive and up-to-date overview of the subject matter and revised the introduction as suggested in the main text.

  1. The quality of the figures should be improved.

Thank you for your suggestions, we have thoroughly revised these figures as suggested.

  1. The English language should be checked carefully.

Thank you for your suggestions. We have carefully proofread and edited the entire text to improve the quality of the English language and ensure clarity and coherence throughout the manuscript.

  1. It will be good if the authors can add molecular dynamics simulation.

We value your suggestion to include molecular dynamics simulations in our study. While we recognize the importance of molecular dynamics simulations, the computational resources available to us were indeed limited, which posed a significant challenge in carrying out extensive simulations within the given timeframe. Despite the resource constraints, we were able to perform a single molecular docking experiment. We appreciate your understanding of the practical constraints we faced and your valuable feedback.

  1. The conclusion should be added.

Thank you for your suggestions, we have added a conclusion section in the main text.

Reviewer 3 Report

Comments and Suggestions for Authors

The author of the manuscript titled "Network Pharmacology Analysis of the Potential Pharmacological Mechanism of a Sleep Cocktail” through a network approach identified the potential drug targets of five dietary supplements "5-Hydroxytryptophan", "Magnesium", "Vitamin B6", "L-Theanine", "GABA", and "Melatonin" that showed improvement for insomnia in preliminary unpublished experiments. Although the results are interesting, the writing is poor and requires extensive language editing. Here are some examples.

The phrase “…Feb 2024). Import the CanonicalSMILES format files into the SwissTargetPrediction[13] platform (http://www.swisstargetprediction.ch/, accessed on 22nd Feb 2024), set its attribute to "homosapiens" in lines 73, 74 and 75, it needs clearly a word connector, and some words need spaces to make the text clear.

The phare “to better understand these ingredients, Disease oncology (DO) enrichment analysis was performed.” in line164. It needs to specify what is needed to understand about the ingredients. Also, what is disease oncology?

The phrase “This indicates the binding between the active compounds and the protein targets is stable, and the compounds involve Melatonin and 5-HTP” in lines 315-316 it also needs a logical word connector.

The phrase “A binding score of less than -5.0 kcal/mol indicates moderate affinity, and a binding score of less than -7.0 312 kcal/mol indicates high affinity binding”. who defines this threshold? In lines 311 It needs at least a citation or an explanation in the methodology.

The phrase “obtained using PILI” line 322 it should be PLIP.

And so on.

Moreover, the figures are in low resolution, It needs to be improved. Figure 5 had a error in the degree word.

Additionally in the discussion section is only a restatement of the results and the importance of them are missed.

Comments on the Quality of English Language

The phrase “…Feb 2024). Import the CanonicalSMILES format files into the SwissTargetPrediction[13] platform (http://www.swisstargetprediction.ch/, accessed on 22nd Feb 2024), set its attribute to "homosapiens" in lines 73, 74 and 75, it needs clearly a word connector, and some words need spaces to make the text clear.

The phare “to better understand these ingredients, Disease oncology (DO) enrichment analysis was performed.” in line164. It needs to specify what is needed to understand about the ingredients. Also, what is disease oncology?

The phrase “This indicates the binding between the active compounds and the protein targets is stable, and the compounds involve Melatonin and 5-HTP” in lines 315-316 it also needs a logical word connector.

The phrase “A binding score of less than -5.0 kcal/mol indicates moderate affinity, and a binding score of less than -7.0 312 kcal/mol indicates high affinity binding”. who defines this threshold? In lines 311 It needs at least a citation or an explanation in the methodology.

The phrase “obtained using PILI” line 322 it should be PLIP.

And so on.

Author Response

The author of the manuscript titled "Network Pharmacology Analysis of the Potential Pharmacological Mechanism of a Sleep Cocktail” through a network approach identified the potential drug targets of five dietary supplements "5-Hydroxytryptophan", "Magnesium", "Vitamin B6", "L-Theanine", "GABA", and "Melatonin" that showed improvement for insomnia in preliminary unpublished experiments. Although the results are interesting, the writing is poor and requires extensive language editing.

Thank you for your comments. We have carefully proofread and edited the entire text to improve the quality of the English language and ensure clarity and coherence throughout the manuscript. 

Here are some examples.

The phrase “…Feb 2024). Import the CanonicalSMILES format files into the SwissTargetPrediction[13] platform (http://www.swisstargetprediction.ch/, accessed on 22nd Feb 2024), set its attribute to "homosapiens" in lines 73, 74 and 75, it needs clearly a word connector, and some words need spaces to make the text clear.

Thank you for your comments. We have revised the phrase as suggested.

The phare “to better understand these ingredients, Disease oncology (DO) enrichment analysis was performed.” in line164. It needs to specify what is needed to understand about the ingredients. Also, what is disease oncology?

Thank you for your comments. We are really sorry about the typo. It should be “Disease ontology” and and we've updated it as you suggested. “To gain further insight into the disease connections of the ingredients, a disease ontology (DO) enrichment analysis was conducted.” In line 198.

The phrase “This indicates the binding between the active compounds and the protein targets is stable, and the compounds involve Melatonin and 5-HTP” in lines 315-316 it also needs a logical word connector.

Thank you for your comments. We have revised it as suggested

The phrase “A binding score of less than -5.0 kcal/mol indicates moderate affinity, and a binding score of less than -7.0 312 kcal/mol indicates high affinity binding”. who defines this threshold? In lines 311 It needs at least a citation or an explanation in the methodology.

Thank you for your comments. This threshold is just a rule of thumb and we are really sorry for the confusion. We've revised it to “Lower values indicate a stronger binding ability, and binding score less than -5.0 kcal/mol are generally considered to indicate medium affinity, while binding fractions less than -7.0 kcal/mol are considered to indicate high affinity.” You'll find this in line 343.

The phrase “obtained using PILI” line 322 it should be PLIP.

Thank you for your comments. We have revised it as suggested

And so on.

Moreover, the figures are in low resolution, It needs to be improved. Figure 5 had a error in the degree word.

Thank you for your comments. We have revised Figure 5 as suggested and improved the quality of all figures.

Additionally in the discussion section is only a restatement of the results and the importance of them are missed.

Thank you for your comments. We have shortened the statements on results and added more discussion of the significance.

Round 2

Reviewer 1 Report

Comments and Suggestions for Authors

The authors addressed all my concerns and significantly improved the manuscript. Therefore, I now recommend it for acceptance.

Author Response

Thank you for your valuable feedback and recommendation. We are glad that we have addressed your concerns and improved the manuscript to your satisfaction.  We appreciate your time and effort in reviewing the manuscript and providing your thoughtful suggestions.

Reviewer 3 Report

Comments and Suggestions for Authors

The authors have addressed previous observations and improved the manuscript. It is clear the significance of their research. Still have some observations, there are typos in line 133 .. whh…; the explanation for the panel A of figure 11 is need. The figure 3 panel A, it still had poor resolution.

Comments on the Quality of English Language

Still have some observations, there are typos in line 133 .. whh…; the explanation for the panel A of figure 11 is need. The figure 3 panel A, it still had poor resolution.

Author Response

The authors have addressed previous observations and improved the manuscript. It is clear the significance of their research. Still have some observations, there are typos in line 133 .. whh…;

Thank you for your comments. We have revised it as suggested. (line 136)

the explanation for the panel A of figure 11 is needed.

Thank you for your comments.  We have added the explanation for panel A of Figure 11. (line 407)

The figure 3 panel A, it still had poor resolution.

Thank you for your comments.  Converting files to PDF format will cause the image resolution to drop, and we have manually edited Figure 3 of the PDF file to ensure high resolution.